# Structural Analysis of the Menangle Virus P Protein Reveals a Soft Boundary between Ordered and Disordered Regions

**DOI:** 10.3390/v13091737

**Published:** 2021-08-31

**Authors:** Melissa N. Webby, Nicole Herr, Esther M. M. Bulloch, Michael Schmitz, Jeremy R. Keown, David C. Goldstone, Richard L. Kingston

**Affiliations:** 1School of Biological Sciences, University of Auckland, Auckland 1010, New Zealand; melissa.webby@bioch.ox.ac.uk (M.N.W.); nicole.herr@auckland.ac.nz (N.H.); e.bulloch@auckland.ac.nz (E.M.M.B.); jeremy@strubi.ox.ac.uk (J.R.K.); d.goldstone@auckland.ac.nz (D.C.G.); 2Department of Biochemistry, University of Oxford, Oxford OX1 2JD, UK; 3School of Chemical Sciences, University of Auckland, Auckland 1010, New Zealand; m.schmitz@auckland.ac.nz; 4Division of Structural Biology, University of Oxford, Oxford OX1 2JD, UK

**Keywords:** Paramyxoviruses, RNA-dependent RNA polymerase, pararubulaviruses, intrinsically disordered proteins, protein self-association, conformational exchange, small-angle X-ray scattering, analytical ultracentrifugation, solution NMR spectroscopy, X-ray crystallography

## Abstract

The paramyxoviral phosphoprotein (P protein) is the non-catalytic subunit of the viral RNA polymerase, and coordinates many of the molecular interactions required for RNA synthesis. All paramyxoviral P proteins oligomerize via a centrally located coiled-coil that is connected to a downstream binding domain by a dynamic linker. The C-terminal region of the P protein coordinates interactions between the catalytic subunit of the polymerase, and the viral nucleocapsid housing the genomic RNA. The inherent flexibility of the linker is believed to facilitate polymerase translocation. Here we report biophysical and structural characterization of the C-terminal region of the P protein from Menangle virus (MenV), a bat-borne paramyxovirus with zoonotic potential. The MenV P protein is tetrameric but can dissociate into dimers at sub-micromolar protein concentrations. The linker is globally disordered and can be modeled effectively as a worm-like chain. However, NMR analysis suggests very weak local preferences for alpha-helical and extended beta conformation exist within the linker. At the interface between the disordered linker and the structured C-terminal binding domain, a gradual disorder-to-order transition occurs, with X-ray crystallographic analysis revealing a dynamic interfacial structure that wraps the surface of the binding domain.

## 1. Introduction

The Paramyxoviruses are a large family of negative-sense, non-segmented, single-stranded RNA viruses. Paramyxoviruses enter the host via the respiratory tract; infect most vertebrate species; and cause serious disease in both humans and animals. Among the well-established human pathogens in this family are the measles virus, mumps virus, and the four pathogenic human parainfluenza viruses (HPIV1–HPIV4).

The widespread distribution of paramyxoviruses creates the potential for cross-species transmission. Bats are now recognized as a particularly important animal reservoir for paramyxoviruses [1,2]. Pteropodid bats are the natural host for Hendra and Nipah viruses, which have spilled over into human and animal populations numerous times in the past three decades [3,4]. While Hendra and Nipah viruses cause no apparent disease in bats, they are highly pathogenic when transmitted to other animal populations, where they can cause fatal encephalitis [5]. Bats host many other paramyxoviruses with zoonotic potential. Among these is Menangle virus (MenV) [6,7], a pararubulavirus that was first identified following a novel disease outbreak in domesticated pigs [8].

All paramyxoviruses require three proteins to facilitate virally-directed RNA synthesis; one protein to package, protect and organize the RNA genome, and two proteins that together form the viral RNA-dependent RNA polymerase (RdRp). The genome is packaged by the nucleocapsid protein (N protein), resulting in formation of a helical protein-RNA complex termed the nucleocapsid [9,10,11,12,13,14]. The viral RdRp [15,16,17,18] performs transcription and genome replication sequentially, using the nucleocapsid as a template [19]. The large protein (L protein) is the catalytic subunit of the RdRp, facilitating synthesis and chemical modification of RNA, while the phosphoprotein (P protein) is the non-catalytic subunit of the RdRp, enabling its processive translocation along the viral nucleocapsid, and modulating the interactions between the nucleocapsid protein and RNA.

The P protein effectively functions as a flexible hub that coordinates binding activities that are integral to polymerase function. In all paramyxoviruses, the phosphoprotein self-associates via a centrally located coiled-coil, while the regions N-terminal and C-terminal to the coiled-coil are predominantly disordered, and involved in binding both viral and cellular proteins. Structural analysis of the coiled-coil has established that it exists as a tetramer [16,20,21,22,23]. The four helices contributing to the tetramer generally run parallel, giving the coiled-coil four-fold rotational symmetry along its central axis [16,20,21,23]. In the case of the Mumps virus P protein, however, the coiled-coil is an antiparallel “dimer-of-dimers” with two-fold rotational symmetry perpendicular to the central axis [22,24]. As a result, the chain termini of the Mumps P protein are not all juxtaposed. Hence, some variation in the exact mode of oligomerization can clearly be tolerated [16,22,24], though the functional implications are presently unknown. To date, only a single structure of an intact paramyxoviral P/L complex has been determined, from parainfluenza virus 5 [16]. In this case, the C-terminus of the coiled-coil is the point of attachment with the L protein, and the body of the coiled-coil projects out a considerable distance from the catalytic subunit [16].

It is the C-terminal region of the phosphoprotein, encompassing the coiled-coil and downstream sequences, that allows the polymerase to engage with the nucleocapsid, and is critical for processive RNA synthesis. Following the coiled-coil, there is a flexible linker that varies widely in length and composition between paramyxoviruses [25], and then a structured binding domain (the foot domain or X domain) that interacts with both the L protein [16,26] and the RNA-bound N protein [27,28,29,30,31]. In all cases the binding domain forms a compact bundle of three alpha-helices [30,32,33,34,35], though in the rubulaviruses, the binding domains can be unstable, and exist natively in a partially structured state [34,35,36]. Different faces of the three-helix bundle are implicated in binding of L and N [16,26]. The C-terminal binding domains are expected to be spatially adjacent, as a result of tetramerization of the P protein, and the tethering activity of the linker, and likely bind the RNA-associated N protein in a multivalent fashion.

It is not known how the various binding events mediated by the C-terminal region of the P protein are coordinated to facilitate polymerase translocation, the threading of the RNA genome through the active site of the polymerase, and the transient displacement of the nucleocapsid protein from the genome. However, it seems likely that extension and compaction of the dynamic linker must occur to facilitate RdRp movement. Polymerase translocation must involve the repeated attachment and release of the binding domain to the N protein subunits of the nucleocapsid, and possibly also the L protein.

Here we report structural and biophysical analysis of the C-terminal region of the MenV P protein. MenV belongs to the pararubulavirus genus, for which there has been little prior characterization of the P protein. We investigate the oligomerization state of the MenV P protein in solution, as well as the structural characteristics of the flexible linker that connects the coiled-coil and the C-terminal binding domain. The results obtained broaden our understanding of the organization of the paramyxoviral P protein, an essential component of the paramyxoviral replication machinery.

## 2. Materials and Methods

### 2.1. Protein Production and Purification

DNA encoding the Menangle virus P protein (UniProt ID Q91MK1; Genbank Accession AF326114), codon-optimized for expression in *Escherichia coli*, was commercially synthesized (Invitrogen GeneArt, Thermo Fisher Scientific). Bacterial expression plasmids enabling the production of six truncated MenV P proteins (‘CC-L-BD’=P_209-388_; ‘CC-L’=P_209–336_; ‘CC’=P_209–271_; ‘L-BD’=P_267–388_; ‘L’=P_267–328_; ‘BD’=P_337–388_) were generated using standard molecular biology procedures (Appendix A). These constructs appended protease-cleavable N-terminal affinity tags to the P protein to facilitate purification. Additional expression plasmids were generated for the crystallographic analysis (Appendix A). These constructs enabled production of the final part of the MenV P linker and the binding domain, fused to the C-terminus of maltose-binding protein (MBP).

All proteins were produced in *E. coli* BL21(DE3) (Stratagene). Transformed bacteria were cultured in LB media, supplemented with 1% (*v*/*v*) glycerol and selective antibiotics. Cultures were shaken in baffled flasks at 37 °C until the absorbance at 600 nm was ~0.6. Gene expression was induced with the addition of 0.5 mM IPTG, and the cultures subsequently maintained at 18 °C for 18 h, before pelleting the bacteria using centrifugation.

Bacterial cell pellets were subsequently resuspended in a construct-specific lysis buffer (Appendix A), supplemented with protease inhibitors (cOmplete EDTA-Free Protease Inhibitor Cocktail, Roche) and lysed by passage through a cell disruptor. Cellular debris was removed using high speed centrifugation. Proteins were isolated by affinity chromatography, using appropriate resin, wash and elution buffers (Appendix A). Where required, the affinity tag was cleaved by incubation with Tobacco Etch Virus (TEV) protease (4–18 °C for ~18 h), either pre-elution (GST fusion proteins) or post-elution (poly-histidine fusion proteins). Subsequently, ion exchange chromatography and size exclusion chromatography (SEC) were used sequentially to purify all proteins (Appendix A). Proteins in SEC elution buffer (Appendix A) were spin concentrated and final protein concentrations were determined from absorbance at 280 nm, with molar absorption coefficients estimated from amino acid composition [37]. The linker domain (P_267–328_) contained no side chain chromophores absorbing in the near UV, and therefore its concentration was determined using the biuret assay, as previously described [38].

### 2.2. SEC-MALLS

For analysis using size exclusion chromatography coupled to multi-angle laser light scattering (SEC-MALLS), proteins in SEC elution buffer (Appendix A) were loaded at varying concentrations onto either Superdex 75 (L-BD; L; BD) or Superdex 200 (CC-L-BD) SEC columns (Cytiva). The columns were attached to a Dionex HPLC equipped with a PSS SLD7000 7-angle multi-angle laser light scattering detector and Shodex RI-101 differential refractive index detector. The mass averaged molar mass of each sample was determined from the refractive index and light scattering measurements using the PSS WinGPC UniChrom software, under the assumption of Rayleigh scattering. A constant refractive index increment (∂n/∂c) of 0.186 was used to estimate all protein concentrations. The MALLS detector was calibrated using a bovine serum albumin solution (2 mg/mL).

### 2.3. Large Zone SEC

For large zone size exclusion experiments, a Tricon 5/5 Superdex 200 column was equilibrated at ambient temperature (20–22 °C) with a buffer containing 12.5 mM MOPS/KOH and 250 mM NaCl. The column void volume (V_0_) and included volume (V_I_) were determined using blue dextran (fully excluded standard, eluting at V_0_) and tyrosine (fully included standard, eluting at V_0_ + V_I_). Solutions of CC-L-BD at varying concentrations (2–90 μM) were loaded onto the column until a plateau of constant protein concentration was achieved, as monitored by UV absorbance (Appendix A). The midpoint of the leading edge (V_e_) was subsequently determined by numerical integration of the area under the peak [39]. From the midpoint, a mass average partition coefficient ($\bar{\sigma }$) for the protein was calculated:(1)$$\bar{\sigma }=\frac{\left(V_{E}-V_{0}\right)}{V_{I}}$$

The change in mass average partition coefficient with protein concentration was modeled under assumption of a dimer–tetramer equilibrium. In this case, the experimental data should be described by [39,40]:(2)$$\bar{\sigma }=f_{P2}\sigma_{P2}+\left(1-f_{P2}\right)\sigma_{P4}$$
where *σ_P2_* and *σ_P4_* are the partition coefficients of the dimer and tetramer respectively, and f_P2_ is the fraction dimer:(3)$$f_{P2}=\frac{c_{P2}}{c_{P2}+c_{P4}}=\frac{2\left[P2\right]}{2\left[P2\right]+4\left[P4\right]}=\frac{2\left[P2\right]}{\left[P_{Total}\right]}$$
where c_P2_ and c_P4_ are the mass concentrations of the dimer and tetramer; [P2] and [P4] are the molar concentrations of the dimer and tetramer; and [P_total_] is the total molar protein concentration.

Through consideration of mass balance and the chemical equilibrium, the dimer concentration and hence the fraction dimer can be found as the solution to the quadratic equation:(4)$$f_{P2}=\frac{-1+\sqrt{1+4K\left[P_{total}\right]}}{2K\left[P_{total}\right]}$$
where *K* is the equilibrium association constant.

### 2.4. Analytical Ultracentrifugation

All analytical ultracentrifugation (AUC) experiments, on construct CC-L-BD in SEC elution buffer (Appendix A), were conducted at 20 °C in a Beckman Coulter model XL-I centrifuge equipped with absorbance optics. Sample and reference solutions were transferred into quartz-windowed cells with a double-sector centerpiece and loaded into a Beckman Coulter 8-place An-50 Ti Rotor. Sedimentation velocity experiments were carried out at 50,000 rpm, with protein sedimentation followed by measuring absorbance at 275 nm. Data were analyzed with the program SEDFIT to derive sedimentation coefficient distributions c(s), and molar mass distributions c(M) [41,42]. The program SEDNTERP [43] was used to estimate the required values for the buffer viscosity (0.01036 cP), buffer density (1.00949 g/mL), and protein partial specific volume (0.750 mL/g). Sedimentation equilibrium experiments were carried out at three protein concentrations (1 mM, 2 mM, and 4 mM) and two rotor speeds (10,000 and 13,000 rpm). Data were collected at a wavelength of 228 nm, with a step size of 0.001 cm, and averaging of 25 measurements per step. The program SEDPHAT [44,45] was used for comparative model fitting, assuming either a single sedimenting species, or a variety of schemes for reversible self-association (with the molar mass of the monomer as a fixed parameter [46,47]). Models were compared by analysis of the global goodness of fit (chi-squared values) and by inspection of model residuals for systematic deviations.

### 2.5. Small Angle X-ray Scattering (SAXS)

#### 2.5.1. Data Collection and Model Free Analysis

SAXS data (Table 1) on proteins CC-L-BD, L and L-BD in SEC elution buffer (Appendix A) were collected at the Australian synchrotron SAXS/WAXS beamline over a q range 0.01–0.6 Å^−1^ (where q is the amplitude of the momentum transfer vector). Data were collected from protein solutions in a 1.5 mm capillary at a temperature of 10 °C, under continuous flow conditions. The program ScatterBrain (Australian Synchrotron) was used to obtain circularly averaged 1D scattering profiles. Baseline subtraction and model-free data analysis was performed using components of the ATSAS software package [48]. The generation of Guinier and Kratky plots was performed with PRIMUS [49], while calculation of the particle distance distribution function P(r) via indirect Fourier transform was performed with GNOM [50].

The relative scattering intensity at zero angle I(0) was estimated via standard Guinier analysis [51]. The Guinier plots were linear, and I(0) scaled linearly with protein concentration in all cases (Appendix A) suggesting the absence of any significant inter-particle interference. This also established that the tetrameric CC-L-BD does not measurably dissociate over the studied concentration range (8–240 µM), reinforcing the hydrodynamic analysis performed previously.

#### 2.5.2. Fit of a Polymer Physics Model to the Data

In the absence of inter-particle interference, the scattered intensity I(q), normalized by the scattering at zero angle I(0), is equal to the form factor of the scatterer P(q).
(5)$$P\left(q\right)=\frac{I\left(q\right)}{I\left(0\right)}$$

An analytic approximation for the form factor of a Kratky–Porod (worm-like) chain was given by Sharp and Bloomfield [52] and has been used for fitting SAXS data for intrinsically disordered proteins and other polymers [53,54,55,56,57,58]:(6)$$P_{SB}\left(q\right)=\frac{2}{x^{2}}\left(x-1+e^{-x}\right)+\frac{b}{L}(\frac{4}{15}+\frac{7}{15x}-(\frac{11}{15}+\frac{7}{15x})e^{-x})$$
where
$x=\frac{q^{2}Lb}{6}$.

The expression (6) is valid only for an infinitely thin chain. The effects of finite chain thickness can be modeled through inclusion of a corrective term into the form factor [59], involving the mean square radius of gyration of the cross section R_c_^2^: (7)$$P\left(q\right)=P_{SB}exp\left(\frac{-q^{2}R_{c}^{2}}{2}\right)$$

The parameter R_c_ is not independent of the contour length L, the two being related by the volume of the protein [56]. This leads to:(8)$$R_{c}^{2}=\frac{\nu_{P}M}{N_{A}2\pi L}$$
where ν_P_ is the partial specific volume of the protein, M its molar mass, and N_A_ is Avagadro’s constant. Together, equations (5)–(8) constitute a model of the small angle X-ray scattering by a worm-like chain, involving three unknown parameters (I(0), L and b) which can be fit to the experimental data I(q). For SAXS analysis, the protein partial specific volume was estimated using a group additivity scheme [60].

SAXS model fitting was carried out using pro Fit (QuantumSoft), using a combination of Monte Carlo and Levenberg–Marquardt algorithms. Fit of the model was excellent over the q range 0–0.35 Å^−1^, with some systematic divergence between data and model observed if the fit was extended to higher scattering angles.

### 2.6. NMR Spectroscopy

The construct L-BD (MenV P_267–388_) was characterized using solution NMR spectroscopy. Details of sample preparation, and the experimental procedures used for ^1^H, ^13^C and ^15^N NMR chemical shift assignment have been fully detailed in a companion paper [61]. The chemical shifts of MenV P_267–388_ were deposited in the Biological Magnetic Resonance Data Bank (http://www.bmrb.wisc.edu) under the accession number 27634.

Acquisition and processing of 2D ^1^H ^15^N HSQC and 3D ^15^N-edited NOESY-HSQC spectra was previously described [61]. Using a ^15^N labeled L-BD sample in pH 7 phosphate buffer [61], a 2D ARTSY-J experiment [62] was executed to enable estimation of ^3^J_HN-Hα_ couplings. The spectra were acquired on a Bruker Avance 600 MHz spectrometer equipped with an inverse triple-resonance pulsed-field gradient cryoprobe. The interleaved time domain data matrix consists of 2 × 1024* (^1^H, acquisition time 136.4 ms) × 256* (^15^N, acquisition time 29.0 ms) complex data points. The total dephasing time was set to 40 ms. Squared sine bell functions were used to apodize the data in both dimensions, which were also zero filled prior to Fourier transformation, to yield a final data matrix size of 4096 (^1^H) × 1024 (^15^N) real points. Spectral processing was performed using NMRPipe [63], and the quantitation of spectral peak intensities was carried out with PINT [64]. The ^3^J_HN-Hα_ couplings were estimated from signal intensities in the paired ARTSY-J spectra, for which ^3^J_HN-Hα_ dephasing was either omitted, or included, with uncertainties estimated from the noise in the spectral base-plane [62]. Results are presented in Appendix A.

The Vadar webserver [65] was used to derive the torsion angle Φ from crystallographic model coordinates. The associated ^3^J_HN-Hα_ coupling constants were calculated using a Karplus relation, parameterized as follows [66]:(9)JHN−Hα3=7.97cos2ϕ−60°−1.26cosϕ−60°+0.63

### 2.7. X-ray Crystallography

Crystallization experiments were performed using several MBP-MenV P fusion proteins, in which the final part of the MenV P linker and the binding domain were fused to the C-terminus of MBP (Appendix A). Concentrated proteins (120–160 mg/mL in 12.5 mM Tris/HCl pH 8.5, 150 mM NaCl, 5 mM maltose) were subject to crystallization trials [67] at 18 °C using the sitting-drop vapor diffusion method. Large single crystals of MBP-MenV P_329–388_ were obtained in several conditions (Table 2). In preparation for data collection, crystals were harvested using a small fiber loop and transferred into appropriate cryo-protective solutions (Table 2) before being flash frozen by direct immersion in liquid nitrogen. X-ray diffraction data were collected using the screenless oscillation method, using synchrotron and laboratory X-ray sources. Crystals were maintained at 110K in a cold gas stream throughout data collection. Integration and scaling of X-ray diffraction data was carried out with the program HKL2000 [68]. Phases were determined by the method of molecular replacement, using the program Phaser [69], independently positioning MBP and a previously determined structure for the MenV P binding domain (residues 337–388, PDB Accession Code 4KYC) [35]. The programs Coot [70] and REFMAC [71] were used to perform model building and refinement, respectively. Statistics associated with the X-ray diffraction data and atomic models are collated in Table 2.

## 3. Results

To enable biophysical and structural analysis of the MenV P protein C-terminal region (residues 209-388), six truncated variants were bacterially expressed and purified (Figure 1A, Appendix A), representing all contiguous combinations of the coiled-coil (CC), the flexible linker (L), and the binding domain (BD). The boundaries of the coiled-coil domain were inferred from prior crystallographic analysis of the mumps virus phosphoprotein [22]. Two of the variants, encompassing the coiled-coil alone (CC), or the coiled-coil and the linker (CC-L), formed large soluble aggregates, as assessed by size exclusion chromatography (SEC) and dynamic light scattering (DLS) (data not shown). These constructs were excluded from subsequent analysis.

### 3.1. The Coiled-Coil Drives Tetramer Formation

The four tractable truncated variants (CC-L-BD, L-BD, L, BD) were characterized using SEC-MALLS to assess their oligomeric state. The three variants lacking the coiled-coil (L, BD, L-BD) had nearly Gaussian elution profiles (Figure 1B), with neither the profile shape nor elution volume dependent on the protein loading concentration in the μM range. The mass average molar masses were nearly constant across each peak, and consistent with each protein being monomeric. In contrast, the variant carrying the coiled-coil domain (CC-L-BD) had an asymmetric elution profile (Figure 1B). The mass average molar mass at the leading edge of the peak was correspondent with a tetramer but declined significantly across the trailing edge of the peak. Although there was no large shift in the mean elution volume as the protein concentration was varied (μM concentration range), the peak asymmetry is itself characteristic of a protein undergoing rapid and reversible self-association [72]. The SEC-MALLS data therefore suggested that the coiled-coil domain drives formation of a P protein tetramer, which is capable of dissociation under in vitro conditions.

### 3.2. A Dimer–Tetramer Equilibrium Exists at Sub-Micromolar Protein Concentrations

To further investigate self-association of the MenV P protein, sedimentation velocity analytical ultracentrifugation (SV-AUC) experiments were carried out on CC-L-BD. Protein sedimentation was followed by measuring absorbance at 275 nm and the data were analyzed in SEDFIT [41,42] to derive sedimentation coefficient distributions, c(s), and molar mass distributions, c(M), for the protein at varying concentration (3–50 µM).

At all concentrations examined, CC-L-BD migrated as a single species with a mean sedimentation coefficient of ~3 S (Figure 2). At the lowest protein concentrations examined (<=6 μM) some peak broadening is apparent in the c(s) distribution, and there is a very slight reduction in the mass averaged sedimentation coefficient, obtained through peak integration [42]. The molar mass of the single species observed in c(M) distributions (not shown) was 78 kg/mol, consistent with the expected molar mass of the tetramer (78.8 kg/mol). These data establish that at protein concentrations > 3 µM, the MenV P protein exists predominantly in the tetrameric state, but are not conclusive with respect to tetramer dissociation. Hence, several additional experiments were performed.

Initially, large zone SEC analysis [39,40] was carried out on CC-L-BD (Appendix A). In this experiment, large volumes of protein were loaded onto a miniaturized SEC column [73,74], sufficient to establish a “plateau region” of constant protein concentration (2–90 mM) (Appendix A). Under these conditions, the position of the leading boundary of the elution profile accurately reflects the equilibrium distribution of species at the plateau concentration [39,40]. The leading boundary position increased significantly at the lowest plateau protein concentrations tested (2 µM), suggesting that the tetramer begins to dissociate at these concentrations. However, the range of protein concentrations that could be explored was limited by the sensitivity of the HPLC absorbance optics. Hence, it was not possible to meaningfully fit the experimental data to recover the partition coefficients of the species present, nor the associated equilibrium constant. The data are, however, satisfactorily explained by a dimer–tetramer equilibrium model (Equations 2–4) with physically reasonable values for the parameters (Appendix A).

Subsequently, sedimentation equilibrium analytical ultracentrifugation (SE-AUC) experiments were carried out on CC-L-BD. This technique reports on protein mass rather than size, and hence is more straightforward to interpret than large zone SEC. SE-AUC data were collected at two speeds, and three protein concentrations (1 mM, 2 mM and 4 mM), with the equilibrium distribution of the protein in the AUC cell measured via absorbance at 228 nm. Comparative model analysis established that a reversible dimer–tetramer equilibrium provided the best overall description of the data. When the data were fit to this model (Figure 3), the equilibrium dissociation constant (K_D_) was estimated as 58 nM. However confidence surface analysis [75] established that the point estimate is loosely bounded from above. This results from the relatively high protein concentrations employed in the experiment, which were again dictated by the instrumental sensitivity. If a dimer–tetramer equilibrium exists in solution, with K_D_ = 58 nM and a total protein concentration of 1 µM, the fraction dimer would be 0.2. Thus, only the initial stage of tetramer dissociation can be followed experimentally within the micromolar concentration range, which is consistent with the results obtained using all techniques.

### 3.3. The Linker Sequence Can Be Modeled as A Worm-like Chain and Exhibits Only Weak Local Conformational Preferences

The structural properties of the linker that connects the coiled-coil and the binding domain were further investigated. Small angle X-ray scattering (SAXS) data were collected for three of the truncated variants (CC-L-BD, L-BD, L) (Table 1). Model-free analysis of the SAXS data confirms the expected organization of the MenV P protein C-terminal region. The Kratky plot (Figure 4) for the entire C-terminal region (CC-L-BD) has a bell-shaped appearance, consistent with the major scattering contribution coming from the two structured domains (CC and BD) positioned at either end of the chain. In contrast, the Kratky plot for the isolated linker (L) increases monotonically with the scattering angle, which is characteristic of unfolded proteins [76]. The Kratky plot for L-BD has a hybrid appearance, reflecting its nearly equal partition between structured and unstructured regions. From the pair distance distribution function P(r) (Figure 4), the radius of gyration (R_g_) of the linker is estimated as 26.4 Å by numerical integration. For comparison, the radius of gyration of an intrinsically disordered protein of this chain length (62 residues) is predicted to be 22.7 Å, using a power law expression developed for chemically denatured proteins [77]. Hence, the linker is globally disordered.

If the linker were a completely flexible Gaussian chain with no local rigidity, a Kratky plot would be expected to plateau at high scattering angles [78], which it does not (Figure 4). A more realistic model allows for the stiffness of the polypeptide chain. The SAXS data were therefore fit to a polymer physics model for chain scattering, following established approaches [55,56,57,58]. The worm-like chain (Kratky–Porod) model fit to the data (Equations 5–8) allows for local rigidity in the polypeptide backbone and incorporates a correction for finite chain thickness. Despite the apparent complexity of the model, it involves only three independent parameters: the scattered intensity at zero angle I(0), the contour length L (the length of the chain at maximum possible extension) and the statistical length b (which is twice the persistence length, and a measure of local chain rigidity). Although the analytic expression describing the scattering of the worm-like chain (Equation 6) involves approximations which are strictly valid only for long chains (L/b > 10), and at low scattering angles (qb < 3) [53,54], the model fits the data very well over the q range 0.015–0.30 Å^−1^ (Figure 5).

The SAXS analysis therefore confirms that the linker has an extended conformation with no significantly populated tertiary structure. It can be modeled effectively as a worm-like chain, with a mean persistence length of 13.5 Å and contour length of 192 Å (Figure 5). However, this does not establish that the sequence is uniformly disordered across its entire length. Indeed the mean persistence length is at the upper end of the range observed for intrinsically disordered proteins, with values of 9–10 Å expected for a fully disordered protein [57].

Sequence analysis suggests the potential for localized and transient structure formation within the linker. Relative to folded proteins, IDPs are typically depleted in hydrophobic residues, and enriched in charged and polar residues [79,80]. However, this is not true of the MenV P linker sequence. The linker has the mean hydrophobicity (0.48 assessed using the Kyte–Doolittle hydrophobicity scale [81]) expected of a folded and globular protein [79]. A significant complement of hydrophobic residues, distributed throughout the linker sequence, could promote localized hydrophobically-driven collapse of the polypeptide chain.

In addition, the sequence of the linker is not hypervariable across the para and orthorubulaviruses, and contains three conserved motifs (Appendix A). The consensus sequences of these motifs are TTIKIMDPG (aa 267–275 in MenV P); KKxFKEVPVVVSGP (aa 286–299 in MenV P) and IxLDELARP (aa 312–320 in MenV P), where x is any amino acid. These motifs are most likely related to binding activities of the P protein and are potential loci for coupled binding and folding events.

Finally, the linker sequence is predicted to have some local conformational preferences. In particular, application of the GOR IV algorithm [82]—the standard information theoretic method for secondary structure prediction—identifies several regions (Appendix A) with propensity for either alpha-helical conformation (residues 280–289 and 309–317) or extended beta-conformation (residues 267–272 and 294–298).

To experimentally investigate the local structural characteristics of the linker we used solution NMR spectroscopy. A companion paper [61] describes the near complete NMR chemical shift assignment of the construct encompassing both the linker and the binding domain (L-BD). As expected, backbone amide resonances associated with the linker (L) showed very limited chemical shift dispersion in the proton dimension, consistent with global disorder, while those associated with the binding domain (BD) were widely dispersed, consistent with overwhelming population of the folded state [61]. The propensity for structure formation in the linker was assessed using the ncSPC algorithm (Appendix A), based on the weighted deviation of the chemical shifts of five backbone nuclei (^1^Hα, ^13^CO, ^13^Cα, ^13^Cβ, ^15^N) from reference values specific for intrinsically disordered proteins [83]. The results are fairly concordant with secondary structure prediction (Appendix A) and suggest that very weak local structural preferences for extended beta-conformation or alpha-helical conformation do exist, with these preferences alternating along the length of the linker.

A striking feature of the NMR spectra is the very strong temperature-dependent line broadening that occurs in several regions of the linker. Particularly affected are resonances associated with residues 277–288, a region proximal to the coiled-coil predicted to have alpha-helical propensity (Appendix A), and residues 325–333, the region immediately adjacent to the binding domain. In each case the resonances broaden with increasing temperature (5–25 °C). For the region adjacent to the binding domain, the line broadening is so extensive that some resonances can no longer be detected at the higher temperatures (Figure 6). This suggests that temperature-dependent conformational exchange processes are occurring in both these regions, though it is not possible to infer either the timescale or the exact nature of the underlying conformational changes from the line broadening alone [84,85].

Overall, the NMR analysis confirmed that the linker is not uniformly disordered. It contains localized segments with very weak conformational preferences, and there is evidence for conformational exchange occurring in some regions. A comprehensive investigation of polypeptide chain dynamics using nuclear spin relaxation is ongoing.

### 3.4. The Structural Transition between the Disordered Linker and the Ordered Binding Domain Is Not Abrupt

The NMR-based evidence for conformational exchange in the sequence immediately preceding the binding domain prompted us to reinvestigate the 3D structure of this region using X-ray crystallography. A series of constructs were created for crystallographic analysis in which the originally characterized binding domain of MenV P (amino acids 337–388, PDB ID 4KYC [35]) was N-terminally extended by 7, 8, 10, or 16 residues. As in the prior crystallographic analysis, each of these proteins was fused to a non-cleavable, modified MBP tag, optimized for crystallization [87]. For one of these proteins (MBP-P_329–388_), crystals were obtained in multiple conditions and used for crystallographic structure determination (Table 2).

X-ray diffraction data were collected to ~1.3 Å resolution for crystal form one (space group P2_1_; two molecules in the asymmetric unit) and 1.55 Å resolution for crystal form two (space group P2_1_2_1_2_1_; one molecule in the asymmetric unit). The structures were determined by the method of molecular replacement, providing a total of three instances of the molecule. In each case, helix α1 of the BD is N-terminally extended beyond Ser339, the previously identified N-terminal residue of the domain (Figure 7). Helix α1 is capped at its N-terminus by Pro335, which orients the preceding residues (329–334) such that they form a clasp or latch that runs across the top of the three-helix bundle. Superimposition of the three molecular models shows that this latch is equivalently positioned in each case with only minor variation in the side-chain hydrogen bonding that mediates formation of the latch. In particular, Arg370 and Asn371, located within the loop connecting helices α2 and α3 of the BD, interact with Ser330 and Ser331 in the latch sequence, through their side chain guanidinium and carboxamide groups (Figure 7).

The electron density associated with the residues within the latch is unambiguous (Figure 7), but relatively weak, and this is reflected in the atomic displacement parameters of the structural models (Figure 8). In all cases, the main chain isotropic B-factors of residues within the latch are significantly higher than values in the bordering helices of MBP and the BD, suggesting that the structure is quite dynamic.

### 3.5. Crystallographic Model for the Interfacial Region Is Consistent with the NMR Observations

The structure observed crystallographically at the N-terminal boundary of the binding domain is consistent with the NMR observations. At the beginning of helix α1, the characteristic *i+3* and *i+4* hydrogen bonding interactions of the alpha helix bring the backbone amide group of Ser334 in close proximity to the side chain methyl group of Ala337 (Figure 7). The existence of this close contact in the solution structure was confirmed by detection of the correspondent Nuclear Overhauser Effect (NOE) (Appendix A). The presence of a canonical capping structure on helix α1, involving the side chain hydroxyl group of Ser 334, and the backbone amide group of Ala 337 (Figure 7) is also robustly predicted from the NMR chemical shifts alone, using the MICS algorithm [90]. Hence, the NMR data clearly establishes the existence of the N-terminal cap on the first helix of the binding domain, directing the polypeptide chain across the top of the binding domain, as observed in the crystal structures (Figure 7).

Crystallographic analysis suggests the latch wrapping the surface of the binding domain is likely to be highly mobile (Figure 8), and this is supported by the solution NMR data. In addition to the severe temperature-dependent line broadening of the resonances in this region (Figure 6), suggestive of conformational exchange, we were unable to detect any NOES between the residues in the latch and the body of the binding domain, despite the close contacts observed in the crystallographic models. A caveat is that the most diagnostic NOES in this region involve the rapidly exchangeable protons from the guanidinium group of Arg370 (Figure 7).

However, measurement of ^3^J_HN-Hα_ coupling constants provides additional evidence that the latch residues are undergoing conformational averaging in solution. The ^3^J_HN-Hα_ coupling constants can be quantitatively related to the backbone torsion angle φ via a Karplus-type relation [91], and were measured experimentally for almost all residues in L-BD (Appendix A). For residues within the helices of the structured binding domain, variance between the experimentally measured ^3^J_HN-Hα_ values, and those calculated from the crystal structure using the Karplus relation is small (RMSD 0.99 Hz; calculations performed for chain A, space group P2_1_) However, for the residues located in the latch (residues 329–334), the variance between experimental ^3^J_HN-Hα_ values and those calculated from the crystal structure is much larger (RMSD 2.53 Hz), supporting a high degree of intra-chain mobility in this region. Overall, we hypothesize that the conformation of the latch observed in the crystals is only partially populated in solution, and in exchange with the unstructured state. This hypothesis is consistent with the solution NMR data, though not strictly established by it.

## 4. Discussion

The paramyxoviral P protein is an integral component of the viral RdRp and its C-terminal region enables the RdRp to engage with the viral nucleocapsid. We have investigated the biophysical and structural characteristics of the C-terminal region of the P protein from Menangle virus, a bat borne pararubulavirus. The major focus of our study was to establish the oligomerization state of the MenV P protein, and the structural characteristics of the flexible linker that connects the centrally located coiled-coil to the C-terminal binding domain.

Experimental analysis confirms that the coiled-coil alone drives tetramerization of the MenV P protein (Figure 1 and Figure 2). However, the MenV P protein is not a constitutive tetramer in solution. Using a combination of SEC-MALLS (Figure 1), large zone SEC (Appendix A) and sedimentation velocity and equilibrium AUC (Figure 2 and Figure 3), we showed that as the total protein concentration drops into the nano-molar range, the tetramer begins to dissociate into dimers. From the sedimentation equilibrium AUC experiments (Figure 3), we estimate the dissociation constant K_D_ for the dimer–tetramer equilibrium as ~58 nM. These data suggest that the MenV P protein will likely function as an obligate tetramer in vivo, especially given that RNA synthesis appears to take place in inclusion bodies [92,93,94,95,96], where the local concentration of the P protein would be very high.

Nonetheless, the existence of a measurable dimer–tetramer equilibrium has some structural implications. It suggests that the tetrameric coiled-coil of the MenV P protein is unlikely to be arranged in a parallel configuration with 4-fold rotational symmetry, like the majority of paramyxoviral P proteins characterized to date [16,20,21,23]. Instead, the MenV P coiled-coil is more likely a parallel or antiparallel “dimer-of-dimers”, with only 2-fold rotational symmetry, as this allows for dimer dissociation. The mumps virus P protein is configured in this way [22,24] and its self-associative behavior was previously characterized using sedimentation velocity AUC experiments [22]. At a fixed concentration of 1 mg/mL (~70 μM), only a single tetrameric species was apparent, however that is consistent with our own results for the MenV P CC domain at similar concentrations (Figure 2).

There has been prior characterization of the linker region from the P proteins of Sendai virus (genus *Respirovirus*) [97,98], Nipah virus (genus *Henipavirus*) [23] and measles virus (genus *Morbillivirus*) [99] using NMR spectroscopy, sometimes in combination with SAXS. In all cases the linker is globally disordered. For Sendai virus, the C-terminal half of the ~87 residue linker was analyzed, and was found to be highly flexible, with local conformational sampling effectively dictated by the amino acid type at each position in the linker. These findings were based on both analysis of structural preferences using residual dipolar couplings [97] and analysis of chain dynamics using nuclear spin relaxation [98]. In the case of Nipah Virus, NMR chemical shift analysis suggested that there are weak local preferences for α-helical conformation within the central region of the ~70 amino acid linker [23]. For measles virus, ensemble modeling based on NMR chemical shifts suggested that the ~80 amino acid linker has two non-contiguous regions with weak preferences for α-helical conformation, but is otherwise largely devoid of significantly populated secondary structure [99].

In Menangle virus, the ~62 residue linker is also globally disordered and lacks any tertiary structure. This is readily apparent from model-free analysis of the SAXS data (Figure 4). Fit of the SAXS data to a worm-like chain model (Figure 5) returns a mean persistence length of 13.5 nm for the linker, slightly larger than the value (9–10 Å) expected of a fully disordered protein [57]. This may reflect either the weak secondary structural preferences indicated by NMR chemical shift measurements (Appendix A), and/or the chain-stiffening effects of the six proline residues that are distributed throughout the linker. The contour length of the linker (192 Å) clearly allows for multivalent attachment of the binding domains of the P protein to the nucleocapsid (the inter-subunit distances within the nucleocapsid being of the order of 50 Å [9,10,11,13,14]). However, it is presently unknown if this is required for polymerase function.

Using X-ray crystallography we determined the structure of residues 329–388 of MenV P, encompassing the final part of the linker and the binding domain (Figure 7). The structural analysis showed that helix α1 of the binding domain is N-terminally extended, relative to other paramyxoviral P proteins [30,32,33,34]. Helix α1 is capped at its N-terminus by a proline residue (P335), which directs the polypeptide chain across the top of the three-helix bundle, forming a latch-like structure. This latch is held in place by H-bonding interactions between Arg370 and Asn371, located in a loop connecting α2 and α3 of the NBD, and Ser 330 and 331, located within the latch (Figure 7), though the crystallography suggests a high degree of intrachain mobility in the latch residues (Figure 8). Based on the NMR observations, the latch cannot be the only populated state in solution, and the crystallographic model must capture just one of the conformational states accessible to the protein. Hence, the transition between the disordered linker and the ordered binding domain is not abrupt, and the interfacial region is best described as partially structured. The existence of a partially structured interface between the linker and the C-terminal binding domain has not previously been observed, however this feature may be restricted to MenV and its most immediate clade, as even among the ortho- and para-rubulaviruses, the key residues involved in formation of the latch (Figure 7) are not highly conserved (Appendix A).

Several lines of evidence suggest that the linker region may act as more than a simple tether for the C-terminal binding domains during polymerase translocation. Firstly, in the structure of the parainfluenza virus 5 P/L complex determined by Cryo-EM [16], a single binding domain is found attached to the surface of the L protein, on the face opposite the entrance and exit channels for the template RNA. The binding domain and coiled-coil of the P protein are connected by diffuse yet compact density that is assigned to the linker. While the density was not directly interpretable, the compacted linker appears to wrap the surface of the L protein, and it is possible that the contacts could involve specific sequence motifs within the linker. In a different study, investigating the determinants of liquid–liquid phase separation by the measles virus N and P proteins, very weak binding was detected between residues within the linker, and the RNA bound N protein, which appeared to modulate the physical properties of the phase separated droplets [99]. Hence, it may be that the linker, while intrinsically disordered, and traditionally viewed as a spacer or tether, is the locus for several ancillary binding events during RNA synthesis. Our work has outlined some of the basic physical properties of the C-terminal region of the MenV P protein, which will facilitate understanding of such binding events, as well as the extension and compaction of the linker which appears necessary for polymerase function.

## Figures and Tables

**Figure 1 viruses-13-01737-f001:**
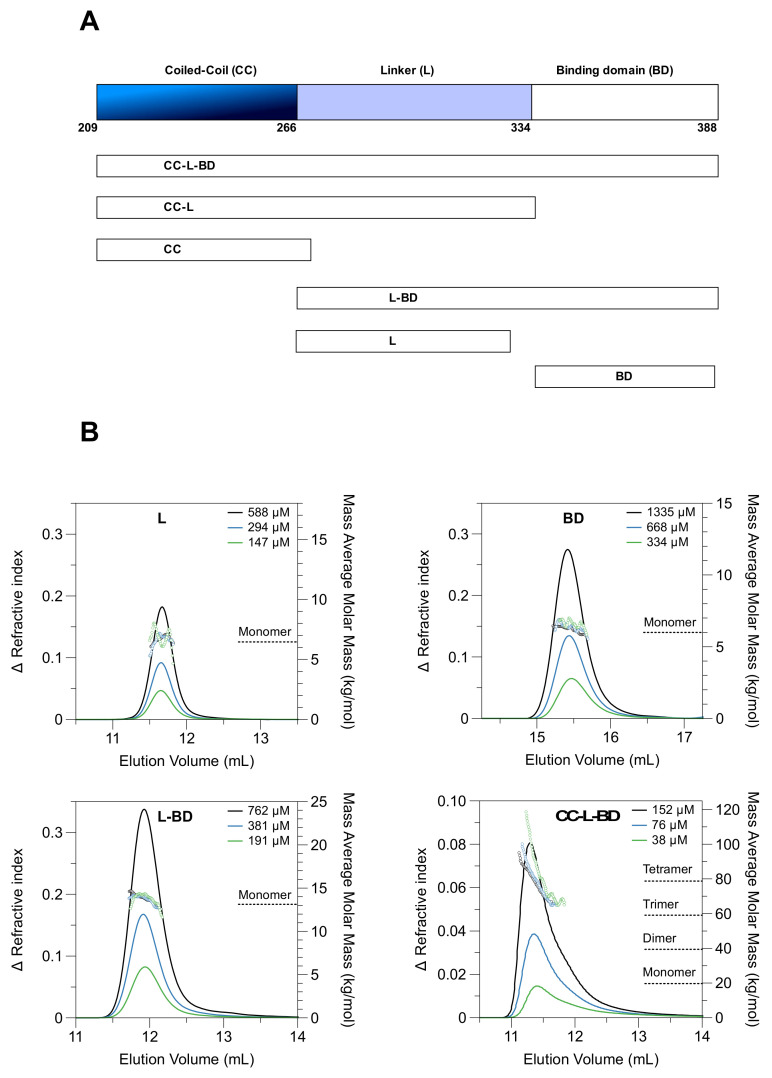
The MenV P coiled-coil drives tetramer formation. (**A**) A schematic showing the truncated proteins employed in this study; (**B**) SEC-MALLS data for four individual proteins: P_267–328_ “L”; P_337–388_ “BD”; P_267–388_ ”L-BD”; P_209–388_ ”CC-L-BD” as indicated. Solid lines show the SEC elution profile (change in refractive index versus elution volume) at the indicated protein loading concentrations. Hollow circles show the mass averaged molar mass estimates resulting from MALLS. Proteins L, BD and L-BD were analyzed using Superdex 75 media (Cytiva), and the plots employ a common vertical scale for refractive index. Protein CC-L-BD was analyzed using Supedex 200 media (Cytiva), and the plot employs a differing vertical scale for refractive index. Molar mass estimates are displayed over the full width at half maximum peak height, in all cases.

**Figure 2 viruses-13-01737-f002:**
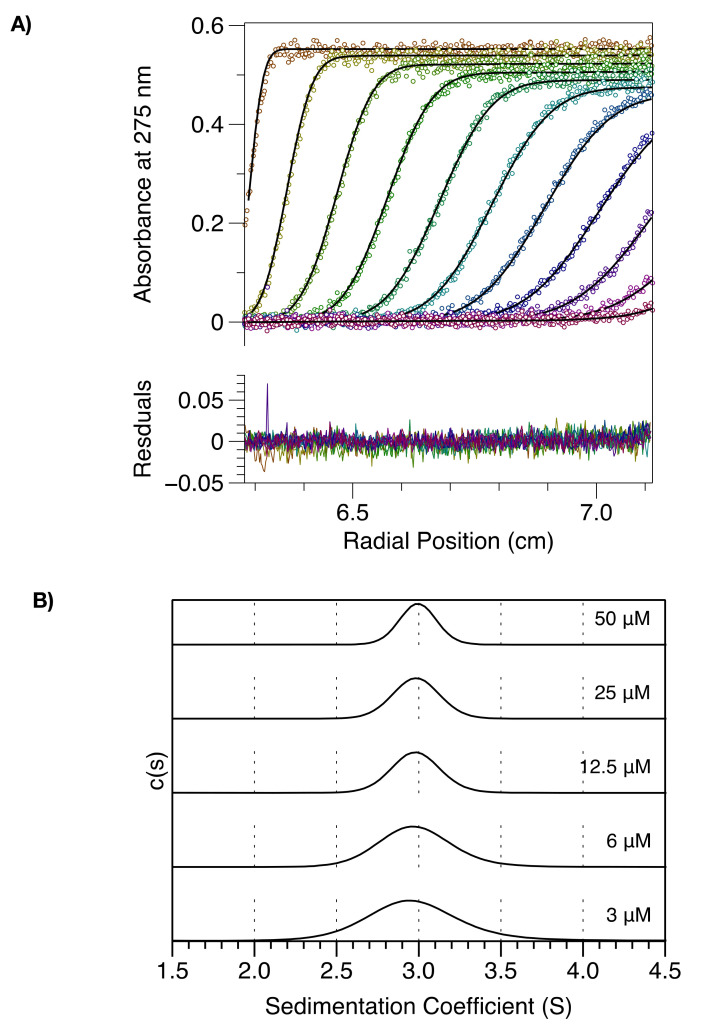
MenV P exists as a tetramer in the micro-molar concentration range. The figure shows sedimentation velocity analytical ultra-centrifugation data and its direct modeling using the program SEDFIT. (**A**) Time-dependent absorbance scans of CC-L-BD (50 μM concentration) undergoing sedimentation (hollow circles), together with the fit model (solid black lines). Every fourth scan is shown, with the model residuals displayed in the bottom panel. (**B**) Sedimentation coefficient distributions, c(s), as a function of protein concentration.

**Figure 3 viruses-13-01737-f003:**
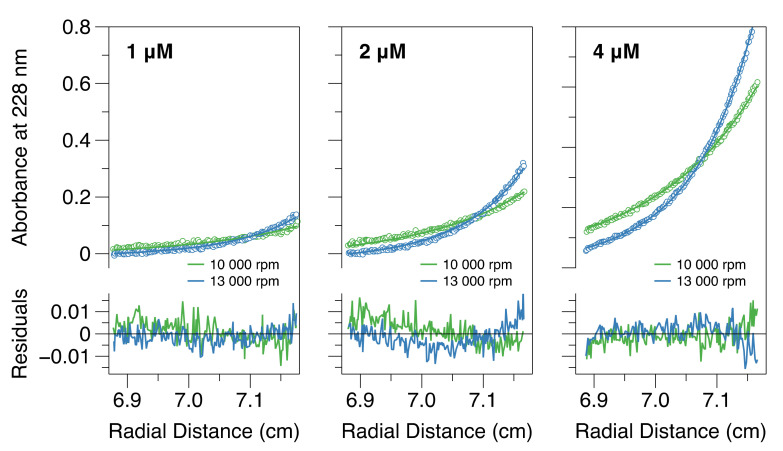
A dimer–tetramer equilibrium becomes apparent at low micromolar protein concentrations. The figure shows sedimentation equilibrium analytical ultra-centrifugation data (hollow circles) for CC-L-BD at three different protein concentrations and two different rotor speeds, and its global fit to a rapid reversible dimer–tetramer self-association model (solid lines). Model residuals are shown below the main plots.

**Figure 4 viruses-13-01737-f004:**
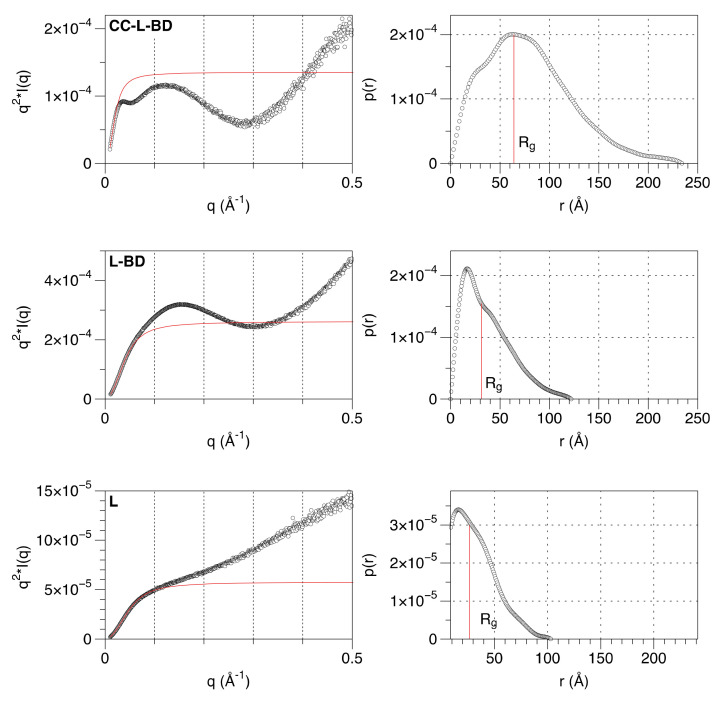
Model-free analysis of SAXS data confirms that the linker is globally disordered. The panels on the left show the Kratky plots (I*q^2^ vs. q) for CC-L-BD (Top), L-BD (middle) and L (bottom). For comparison, the Kratky plot expected of a completely flexible, infinitely thin Gaussian chain, with the same overall radius of gyration (Rg) is shown with a solid red curve. The form factor of the Gaussian chain is given by the Debye function $P\left(q\right)=\frac{2}{x^{2}}\left(x-1+e^{-x}\right)$, where x = (q*R_g_)^2^ [78]. The panels on the right show the correspondent pair distance distribution function P(r). The pair distance distribution functions were calculated by indirect Fourier transformation of the experimental SAXS profiles, as implemented in the program GNOM [50]. The radius of gyration, calculated by numerical integration of the pair distribution functions, is indicated.

**Figure 5 viruses-13-01737-f005:**
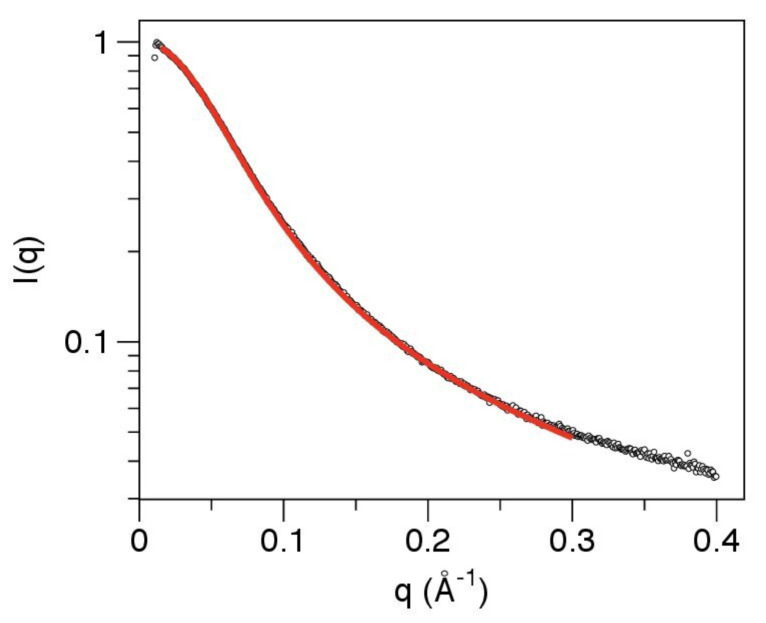
SAXS data for the linker can be fit to a worm-like-chain (Kratky–Porod) model, providing estimates for the contour length and persistence length. Hollow circles show the SAXS data for the linker on a log-linear scale. The red line shows the fit of the model over the q range 0.015–0.30 Å^−1^. Best fit model parameters: contour length 192 Å and persistence length 13.5 Å.

**Figure 6 viruses-13-01737-f006:**
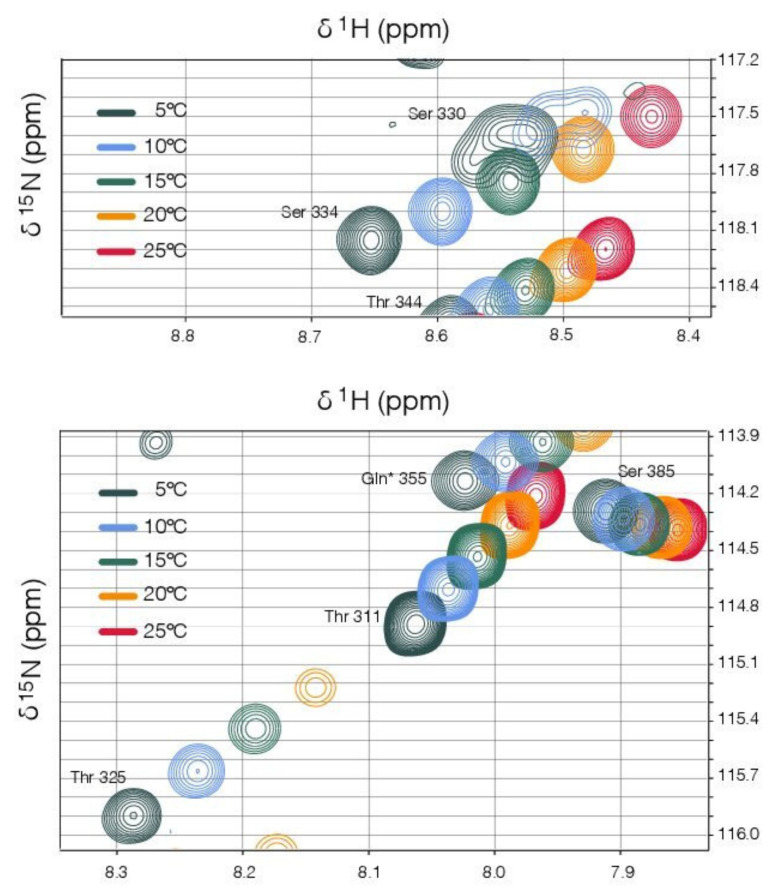
Solution NMR data suggests residues adjacent to the binding domain (residues 325–333) are undergoing temperature-dependent conformational exchange. Selected regions of 2D ^1^H-^15^N HSQC spectra, collected at temperatures between 5 and 25 °C, are overlaid in the two panels, with identical iso-contours displayed. As expected [86], the amide chemical shifts track fairly linearly with temperature in all cases. For the residues T325 (bottom panel) and S330 (top panel), which are typical of this region, the backbone amide peak intensities reduce markedly with temperature, with the S330 resonance broadened beyond the detection limit at temperatures >15 °C.

**Figure 7 viruses-13-01737-f007:**
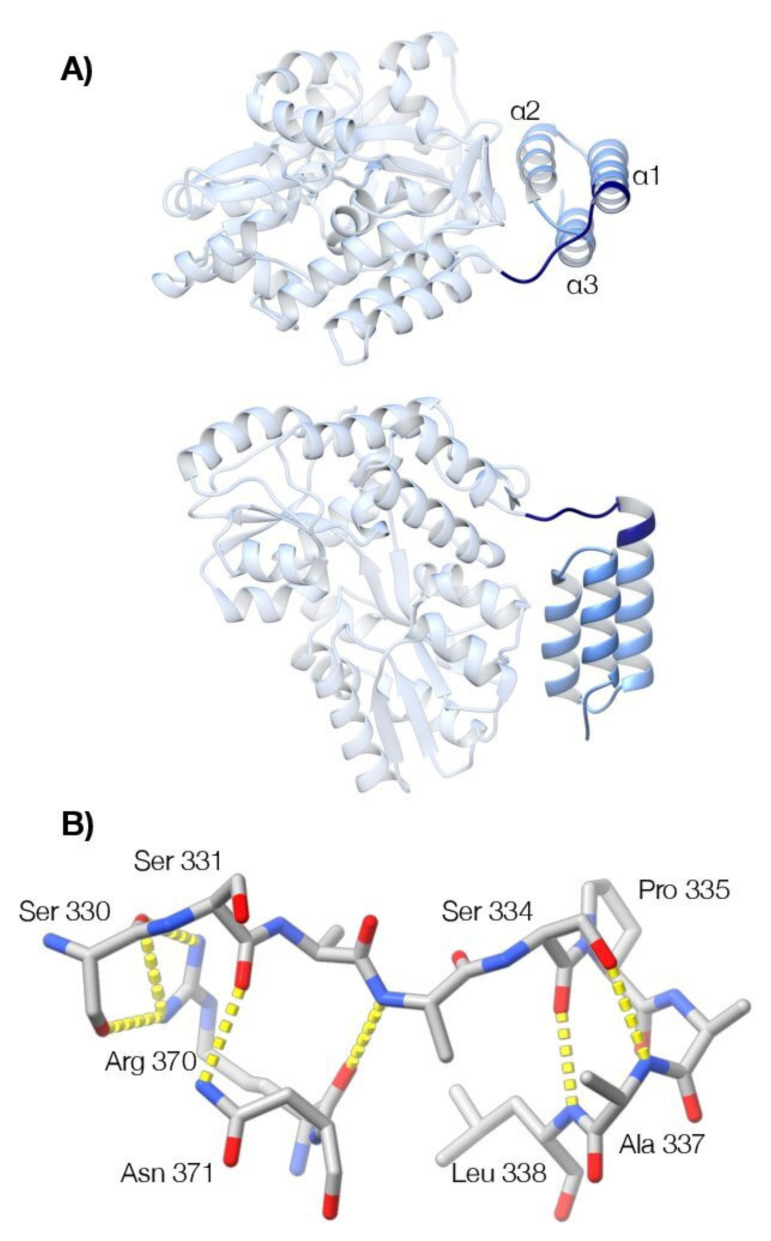
Crystallographic analysis reveals a latch preceding the canonical three-helix bundle of the binding domain. (**A**) The structure of MenV P_329–388_ (blue) fused to MBP (grey), in ribbon representation (P2_1_2_1_2_1_ crystal form, PDB ID 7KD5). Two orthogonal views are shown. The previously determined structure of the binding domain (residues 339–388) is displayed in light blue, while the newly identified structure (residues 329–338) is displayed in navy blue. (**B**) Selected residues that mediate formation of the latch and the N-terminal cap of helix α1 are shown in stick representation (P2_1_2_1_2_1_ crystal form, PDB ID 7KD5). Hydrogen bonding interactions are shown by the yellow dashed lines. The program UCSF ChimeraX was used to prepare the figures [88].

**Figure 8 viruses-13-01737-f008:**
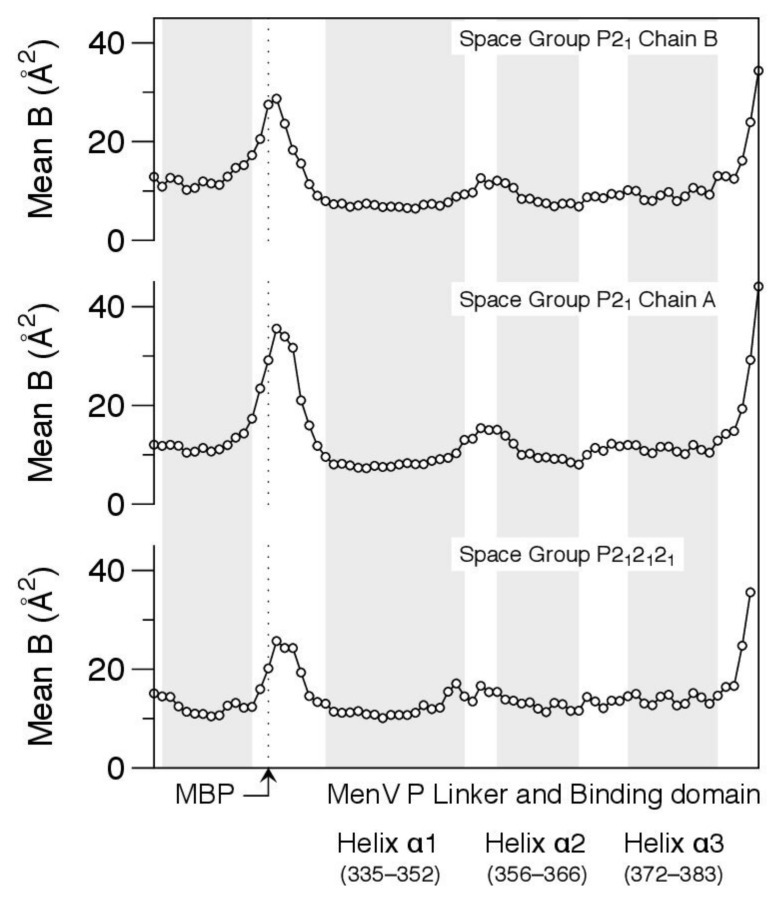
Crystallographic analysis suggests the latch preceding the binding domain is dynamic. Mean isotropic displacement parameters (B-factors) for main chain atoms are shown for the final helix of MBP and MenV P_329–388_, which incorporates the last part of the linker and the binding domain. Positions of the alpha helical secondary structures, as defined by the DSSP algorithm [89], are indicated with gray background shading.

**Table 1 viruses-13-01737-t001:** Small angle X-ray scattering (SAXS) data collection parameters.

		Protein	
	“CC-L-BD” MenV P_209-388_	“L-BD” MenV P_267-388_	“L” MenV P_267-328_
Instrument	Australian Synchrotron SAXS/WAXS beamline, Pilatus 1M detector	Australian Synchrotron SAXS/WAXS beamline, Pilatus 1M detector	Australian Synchrotron SAXS/WAXS beamline, Pilatus 1M detector
Wavelength (Å)	1.0332	1.0332	1.0332
q range (Å^−1^)	0.01–0.60	0.01–0.60	0.01–0.60
Exposure time (s) ^a^	1	2	1
Protein concentration range (g/L)	0.1–4.8	0.35–11.1	0.45–3.62
Buffer composition	12.5 mM MOPS/KOH pH 7.0, 250 mM NaCl	12.5 mM MOPS/KOH pH 7.0, 150 mM NaCl	12.5 mM Tris/HCl pH 8.5, 150 mM NaCl
Temperature (°C)	10	10	10
SASDB accession codes	SASDLG9	SASDLH9	SASDLJ9

^a^ Data were collected under continuous flow conditions.

**Table 2 viruses-13-01737-t002:** MBP-MenV P_329–388_: Protein crystallization conditions and statistics associated with X-ray diffraction data and atomic models.

	Crystal Form I	Crystal Form II
Crystallization Conditions		
Protein concentration (μM)	3300	3300
Crystallization Method	Sitting Drop Vapor diffusion	Sitting Drop Vapor diffusion
Reservoir Solution	1.65 M Ammonium Sulphate 0.20 M Malic acid/KOH pH 5.5	20% (*w*/*v*) PEG 5000 monomethyl ether 0.2 M PIPES/KOH pH 6.7 0.1 M Proline
Temperature (°C)	18	18
X-ray diffraction data		
Cryoprotectant	1.65 M Ammonium Sulphate 0.20 M Malic acid/KOH pH 5.5 1M Lithium Sulphate 5 mM Maltose	20% (*w*/*v*) PEG 5000 monomethyl ether 0.2 M PIPES/KOH pH 6.7 0.1 M Proline 20% (*v*/*v*) Ethlyene glycol 5 mM Maltose
Space group	P2_1_	P2_1_2_1_2_1_
Unit cell dimensions	a = 57.6, b = 70.1, c = 111.7 Å β = 96.5°	a = 67.6 b = 77.4, c = 79.1 Å
X-ray source	Australian Synchrotron Beamline MX1	Rigaku MicroMax-007 HF Rotating Copper Anode
X-ray wavelength (Å)	0.95370	1.54179
Sample Temperature (K)	110	110
Data resolution limits (Å) ^a^	59.26–1.31 (1.33–1.31)	31.24–1.55 (1.57–1.55)
Number of unique observations ^a^	149703 (307)	59991 (2017)
Mean Redundancy ^a^	3.2 (1.5)	6.8 (5.6)
Completeness (%) ^a^	70.7 (2.9)	97.4 (66.5)
R_measure_ ^a^	0.061 (0.857)	0.026 (0.204)
R_merge_ ^a^	0.051 (0.607)	0.026 (0.166)
CC1/2 ^a^	(0.783)	(0.982)
Crystallographic models		
Number of protein molecules in the asymmetric unit	2 × 431 residues	1 × 431 residues
R_work_/R_free_ ^b^	14.9/19.6	16.7/19.0
Total number of protein atoms	6998	3569
Number of water molecules	1321	505
Other ligands	Maltose, Sulphate	Maltose, Ethylene Glycol, PIPES, Proline
Disorder Model	Individual Anisotropic B-factors	Individual Isotropic B-factors
Mean total isotropic B-factor, all protein atoms (Å^2^):	11.7	15.0
Bulk Solvent model	Mask	Mask
RMSD from ideal geometry: Bond lengths (Å)/Bond angles (°)	0.009/1.495	0.010/1.398
Residues in Favoured/Allowed regions of Ramachandran plot (%) ^c^	99/1	99/1
PDB ID	7KD4	7KD5

^a^ Numbers in parentheses are for the highest resolution shell ^b^ Calculated from a randomly selected 5% of observations omitted from all model refinement. ^c^ Defined by the MolProbity web-server.

## Data Availability

The SAXS data have been deposited with the SASBDB (Accession codes SASDLG9, SASDLH9, and SASDLJ9). The crystallographic models and the processed diffraction data have been deposited with the PDB (Accession codes 7KD4 and 7KD5).

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
