# Peer review of "Structural Analysis of the Menangle Virus P Protein Reveals a Soft Boundary between Ordered and Disordered Regions"

_viruses, 2021, doi:10.3390/v13091737_

Round 1

Reviewer 1 Report

The paper nicely describes, how the tetrameric state of the protein is regulated by CC domain.

Using impressive range of biophysical methods not only dominant multimeric state is elucidated, but the equilibrium between dimer and tetramer found. In addition of some minor comments, I would suggest showing data, which are “not shown” to support the conclusions. I would suggest also distinguishing here between flexibility and chemical exchange, as for flexibility NMR relaxation data or something similar would be needed.

Specific comments (not in the order of significance):

Large Zone SEC: some explanation of the method would be helpful. Including “plateau of concentration”, and why it needs to be reached. What is ?? in eq. 1?

Some explanation of each method and why exactly this method was used would help a reader

What about Kratky plot of worm-like model?

“the upper end of the values typically observed for intrinsically disordered proteins” would be nice to state this range, given in the discussion, but could be in the results as well

“The linker has the mean hydrophobicity expected of a folded and globular protein [79], assessed using the Kyte-Doolittle hydrophobicity scale [81]” give the value and expected range

“multiple sequence alignment not shown” show the alignment in the supplementary, to proof the point about conservation

Introduce ncSPC algorithm and when it is mostly used (likely for intrinsically disordered proteins)

Figure 6 data should be shown for all residues at least between 325 and 333, better for all regions were propensity to form secondary structure was reported. BD starts from 329?  Then, only 325-328 are in linker?

The BD definition in X-ray section is in odd with the first paragraph of results.

Why NMR data not used to show flexibility of latch (not just chemical exchange)? NOEs should be shown and not just mentioned, prediction of terminal cap should be elaborated. Right now, it is nothing shown about confirmation of x-ray structure in solution. In this case it seems like it is not the flexibility, but the chemical exchange between ordered/disordered states, as discussed later.

How the J-couplings were calculated for residues outside crystal structure (Linker region) presented in Table S4

Reviewer 2 Report

In this manuscript, the authors used NMR spectroscopy/x-ray crystallography and different biochemical/biophysical methods to study the biophysical and structural features of the C-terminal region of the P protein from Menangle virus. Results from the biochemical/biophysical analyses revealed that the MenV P protein is a tetramer and the coiled-coil domain is responsible for the oligomerization. Although the tetramer can dissociate into dimers at sub-micromolar protein concentrations, the MenV P protein is likely to be in the state of tetramer in vivo given the expected high concentration of the P protein. Biophysical and NMR analysis of the linker that connects the coiled-coil domain and the binding domain, and the x-ray crystallographic studies of the final part of the linker immediately preceding the binding domain and the binding domain revealed that, like the linker region from other viruses, the ~62 residue linker in Menangle virus is also globally disordered. However, the region is not uniformly disordered, contains localized segments with weak conformational preferences, and is structurally dynamic.  Based on the data, the authors discussed the potential functions of the linker region of the MenV P protein. The experimental procedures were adequately described and the experimental data were well presented/interpreted/discussed. The results will be helpful for understanding the biophysical/structural features and the potential functions of the linker region of the MenV P protein.

Minor issues:

  1. It would be better to schematically show the domain structure and the truncated versions they used (start and end positions of the residues) of the MenV P protein as the first part of 1.
  2. 1: Fig. 1A-D were described in the main text but the panels in Fig. 1 were not labeled with A-D.
